Methods

# BiFCo: visualizing cohesin assembly/disassembly cycle in living cells

Emilio González-Martín, Juan Jiménez, Víctor A Tallada

**Cohesin is a highly conserved, ring-shaped protein complex found in all eukaryotes. It consists of at least two structural maintenance of chromosomes (SMC) proteins, SMC1 and SMC3 in humans (Psm1 and Psm3 in fission yeast), and the kleisin RAD21 (Rad21 in fission yeast). Mutations in its components or regulators can lead to genetic syndromes, known as cohesinopathies, and various types of cancer. Studies in several organisms have shown that only a small fraction of each subunit assembles into complexes, making it difficult to investigate dynamic chromatin loading and unloading using fluorescent fusions in vivo because of excess soluble components. In this study, we introduce bimolecular fluorescent cohesin (BiFCo), based on bimolecular fluorescent complementation in the fission yeast *Schizosaccharomyces pombe*. BiFCo selectively excludes signals from individual proteins, enabling the monitoring of complex assembly and disassembly within a physiological context throughout the entire cell cycle in living cells. This versatile system can be expanded and adapted for various genetic backgrounds and other eukaryotic models, including human cells.**

## Introduction

The cohesin complex has long been the focus of significant attention because of its tight association with eukaryotic genomes' structure, function, and segregation. Its interaction with chromatin is critical for loop–extrusion-mediated topologically associated domains, known as TADs, that regulate gene expression over long distances in many cases (Wendt & Peters, 2009; Wutz et al, 2017). Furthermore, it is necessary for preserving the fidelity of other intrinsic DNA processes, such as replication and repair, transcription, condensation, and cohesion, that are essential for maintaining genomic stability (Merkenschlager & Nora, 2016; Osadska et al, 2022). Mutations in some of its components or regulators cause genetic syndromes, known as cohesinopathies, and several types of cancers in humans (reviewed in De Koninck and Losada [2016] and Di Nardo et al [2022]).

The cohesin complex is a ring-shaped structure highly conserved in the evolution from yeast to humans. It is composed of at least two structural maintenance chromosome (SMC) proteins: SMC1 and SMC3 (Psm1 and Psm3 in fission yeast), and the kleisin MCD1/SCC1/RAD21 (Rad21 in fission yeast) (Peters et al, 2008; Yatskevich et al, 2019). When assembled, each SMC protein folds back on itself through a domain known as the "hinge," bringing its N-terminus and C-terminus together to form a domain known as the "head." The hinge domains of both proteins interact, and the α-kleisin Rad21 links the head domains together to close the ring (Haering et al, 2002; Gruber et al, 2003) (Fig 1A).

The proper functioning of cohesin depends largely on the timing of its association/dissociation dynamics from chromatin in tight coordination with the progress of the cell cycle. It starts to be recruited to DNA in the G1 phase and progressively accumulates through the end of the S phase, establishing its cohesive action during DNA replication (Guacci et al, 1997; Michaelis et al, 1997; Uhlmann & Nasmyth, 1998; Bernard et al, 2008). Subsequently, removing it promptly from mitotic chromosomes in the metaphase/anaphase transition is essential to allow the physical separation of sister chromatids (Guacci et al, 1997; Michaelis et al, 1997).

In vertebrate cells, the release from DNA occurs in two steps. The bulk of chromosomal arms cohesin is removed in prophase (in a series of events called the prophase pathway), but centromeric cohesin remains associated until the metaphase to anaphase transition (Losada et al, 1998, 2002; Waizenegger et al, 2000). Centromeric cohesin is then released by a highly conserved mechanism, finely controlled by mitotic regulators, leading to the proteolysis of the Rad21 kleisin by a separase (Cut1sp/ESP1sc/ESPL1hs). Otherwise, separase is kept inactive during most of the cell cycle by its physical association with another protein called securin (Cut2sp/PDS1sc/PTTG1hs). When the mitotic spindle properly captures the chromosomes in the metaphase plate, the anaphase-promoting complex (APC/C) ubiquitinates and targets securin for degradation. This allows the activation of separase, which cleaves Rad21, leading to faithful mitosis in many organisms, including yeasts, xenopus, mice, and humans (Funabiki et al, 1996; Uhlmann et al, 1999; Uhlmann et al, 2000; Hauf et al, 2001). Thus, Rad21 cleavage is essential for ensuring genomic stability.

Centro Andaluz de Biología del Desarrollo, Universidad Pablo de Olavide-Consejo Superior de Investigaciones Científicas-Junta de Andalucía, Seville, Spain

Correspondence: jjimmar@upo.es; valvtal@upo.es

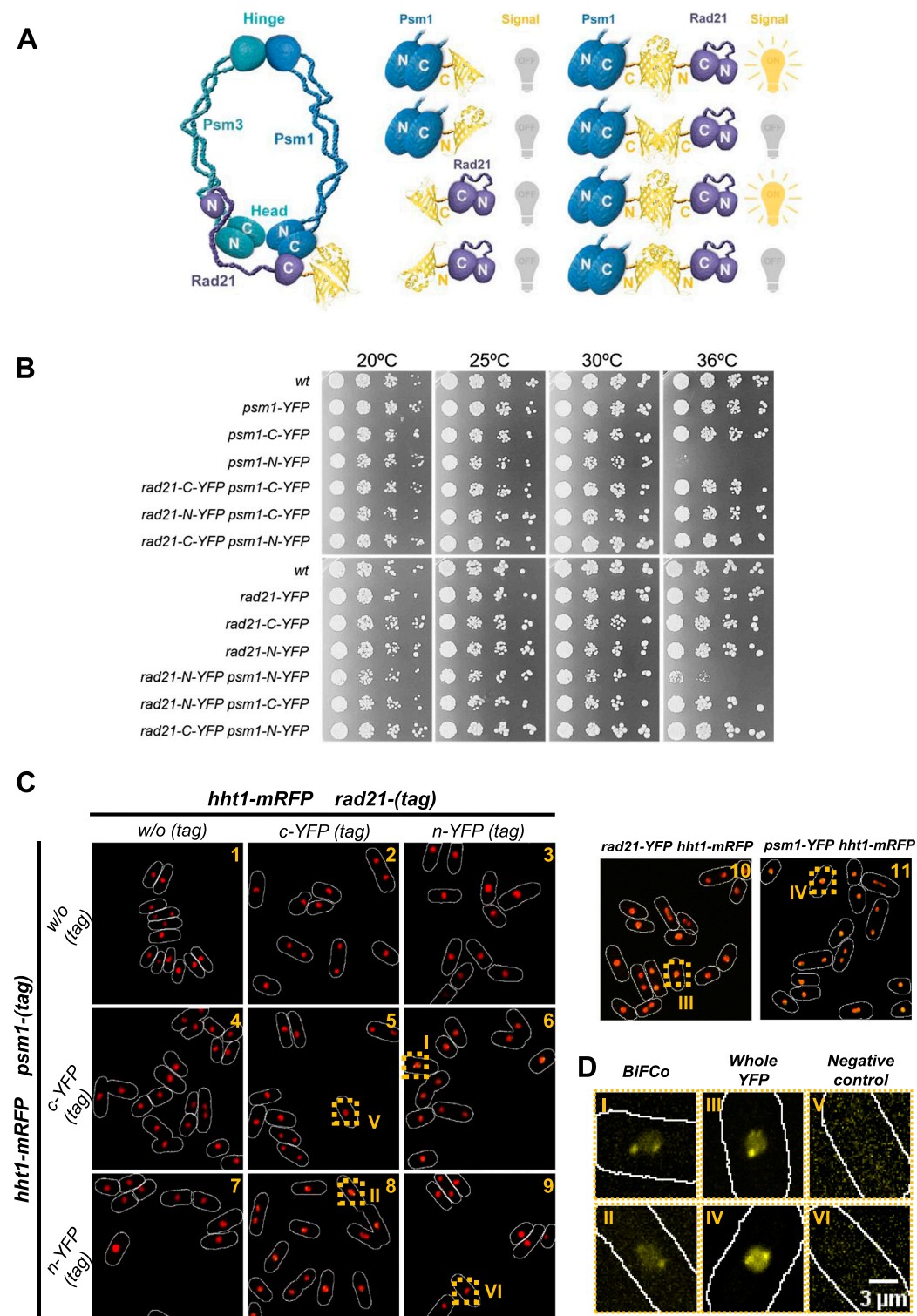

**Figure 1. Bimolecular fluorescence complementation into the cohesin complex.**
**(A)** Cohesin complex cartoon that shows where YFP moieties are fused (left). In that configuration, only indicated combinations are expected to complement and recover fluorescence capacity (right). **(B)** Viability assay. All strains, either single- or double-tagged, along with a wild-type untagged control, were grown to the log phase. The number of cells/ml was scored in a Neubauer chamber to standardize the number of cells of each strain. Fivefold dilutions were spotted at indicated temperatures to assess viability. **(C)** All generated strains represented in A were crossed into the histone Htt1-mRFP strain to co-localize the BiFCo signal within the chromatin. Micrographs presented are sum projections of 21 slices with a Z-step of 0.2 $\mu$m. Only whole YFP fusions (upper right panels 10 and 11) and YFP complementing combinations (panels 6 and 8) give the detectable yellow signals. Scale bar: 8 $\mu$m. **(D)** Representative insets enlargements from section B of the YFP channel alone (roman numbers labels). BiFCo signal (I and II) along with positive (III and IV) and negative (V and VI) controls are shown. Scale bar: 3 $\mu$m.

Remarkably, biochemical analyses show that only a small fraction of cohesin's proteins are associated with chromatin before anaphase (Losada et al, 1998; Schmiesing et al, 1998; Darwiche et al, 1999; Tóth et al, 1999; Tomonaga et al, 2000; Weitzer et al, 2003). This is usually not a significant concern for most in vitro studies, as the soluble fraction is washed away when breaking the cells. However, the high amount of each component in the soluble fraction in intact cells hampers dynamic studies by fluorescent tagging in vivo.

In this work, we developed a system in fission yeast called bimolecular fluorescent cohesin (BiFCo), based on bimolecular fluorescent complementation that enables us to monitor only the fraction of cohesin proteins physically associated with each other in living cells. Bimolecular fluorescent complementation involves fusing two parts of a fluorescent protein (FP) into two cellular proteins. Each part of the FP is unable to emit fluorescence on its own. However, if the proteins of interest physically interact, the FP halves complement in trans and fluorescence emission is restored (Nagai et al, 2001; Hu et al, 2002). This system has effectively been proved in fission yeast for monitoring protein–protein interactions in living cells (Akman & MacNeill, 2009; Grallert et al, 2013). In this study, we fused the two parts of Venus YFP (n-YFP and c-YFP) to the C-terminal ends of Psm1 and Rad21, respectively. This system allows us to monitor the in vivo dynamics of cohesin loading and unloading at different stages of the cell cycle and different regions of chromosomes.

# Results

## Bimolecular fluorescence complementation at the cohesin head domain

It is well established that the C-terminal ends of Rad21 and Psm1 bind to each other close to Psm3 to form the cohesin head domain (Haering et al, 2002; Haering et al, 2004). Therefore, we chose these ends for gene tagging at native loci with the bimolecular complementation system, as it is highly likely that the YFP parts will come close enough in space when these proteins interact to form the ring. In addition, we kept the promoter region of these genes undisturbed to maintain wild-type expression regulation of the tagged genes. We tagged the C-termini of both Rad21 and Psm1, respectively, with the two YFP parts to generate all combinatory by mating and tetrads pulling. These include the reciprocal experimental setup and negative controls (Fig 1A).

We first tested the viability of all strains within the standard growth temperature range of fission yeast wild-type and temperature conditional mutants between 20°C and 36°C, as compared with the whole YFP control taggings and wild-type untagged control (Fig 1B). All strains grew like the wild-type at most temperatures. Only the fusion of Psm1 with the N-terminal moiety of YFP seemed to affect cell viability, albeit only at 36°C. Viability, however, was fully restored when the complementary cYFP part was present in the Rad21 counterpart (experimental configuration) (Fig 1B).

We then tested the fluorescent signal obtained in live cells in the experimental combinations and the respective negative controls. As expected, no signal was obtained in strains carrying only one part or two equal parts of YFP and in the untagged control (Fig 1C: Panels 1, 2, 3, 4, 5, 7, 9 and Fig 1D: Panels V and VI). Fluorescent signal was only observed in the two combinations carrying complementing parts of YFP (Rad21-nYFP, Psm1-cYFP and Rad21-cYFP, Psm1-nYFP) (Fig 1C: Panels 6 and 8 and Fig 1D: Panels I and II), and in the positive controls of proteins carrying a traditional fusion with the whole YFP (Fig 1C: Panels 10 and 11 and Fig 1D: Panels III and IV).

However, although the signal of the complete fusions was evident in both the nucleoplasm and specific foci, the BiFCo signal was much fainter in the nucleoplasm and concentrated mainly on specific foci (see the "subnuclear locations" epigraph below). The fluorescence intensity difference suggests that BiFCo is effectively able to exclusively light up the associated fraction within the complex over the total individual components within a living cell.

## Dynamic analysis of assembled cohesin

The function of the cohesin complex is essential for the correct segregation of sister chromatids, DNA replication, the regulation of gene expression, the generation of topologically associated domains, and ultimately for the maintenance of genomic stability (Nasmyth & Haering, 2005; Losada, 2008; Peters et al, 2008). This highly conserved dynamics in eukaryotes is achieved by regulating the loading and unloading of cohesin on chromatin. The release of sister chromatids to be separated to opposite poles in mitosis is achieved by the degradation of Rad21 kleisin through the action of the Cut1 protease (separase). We, therefore, asked whether the decay of the BiFCo signal could monitor the proteolysis of Rad21 in time-lapse experiments in living cells (Fig 2A). To graphically represent these dynamics, we quantified the nuclear fluorescence at every time-lapse in dividing nuclei and normalized it by the fluorescence of the time-lapse in which we observed the characteristic oblong shape of the nucleus at the beginning of anaphase B (labeled as time 0 in the figures). As shown, there was an abrupt drop of the signal at an interval of 6 min before and 20 min after the normalization point and simultaneous recovery of fluorescence in the two daughter nuclei corresponding to de novo loading in G1 (Fig 2C). These data show that the proteolytic breakdown and reassociation of Rad21–Psm1 interaction can be timely followed in living cells. However, even at the lowest level of fluorescence, a faint nuclear signal is still detected above the background noise. This signal might correspond to assembled free or non-cohesive rings (Huis in 't Veld et al, 2014) or single-chromatid-loaded rings, whose Rad21–Psm1 interaction may not be dislodged in mitosis (Huang et al, 2005; Schmidt et al, 2009; Nasmyth, 2011; Eng et al, 2015).

Next, we tested the robustness of the BiFCo system over time by following two mitotic divisions (Fig 2B). To gather a significant number of nuclei going through two mitoses in the same field of view and to minimize signal variability because of bleaching effects and phototoxicity, we synchronized cells using the conditional cdc25-22 mutant background (Russell & Nurse, 1986). This mutant arrests the cell in late G2 at a high temperature (36°C), but it enters mitosis normally when released at the permissive temperature of 25°C under the microscope. Fig 2D and Video 1 show that the BiFCo signal can be followed for at least two Rad21 cleavage cycles, indicating that cohesin dynamics can be monitored for a long period (8 h) and for more than one assembly–disassembly cycle.

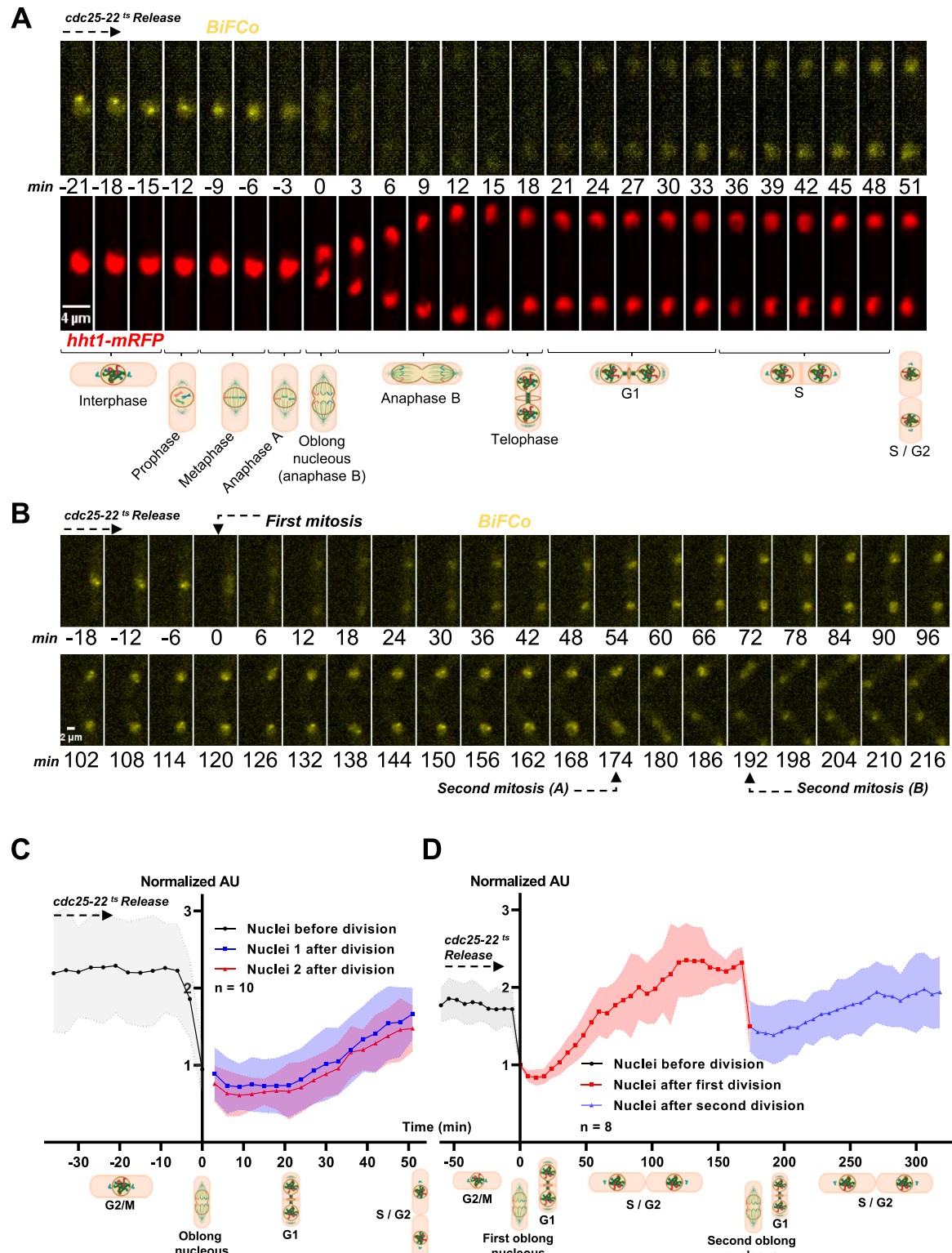

**Figure 2. Cohesin's cycle dynamics.**
**(A)** Individual cells bearing the BiFCo system into the Htt1-mRFP background (as in Fig 1B panel 8) were followed in 3-min time-lapse experiments in the transition from the G2 to G1/S phase to assess predicted degradation in mitosis. Sum projections frames are shown above the schematic representation of cell cycle phases. Scale bar: 4 μm. **(B)** Synchronized *cdc25-22* BiFCo cells were arrested for 4 h at a restrictive temperature and were released and monitored under the microscope for 8 h at a permissive temperature to assess signal recovery and decay over two nuclear divisions. Scale bar: 2 μm. **(C)** Fluorescent signal quantification over time. Ten individual nuclei divisions from section A were analyzed, representing average fluorescence at each time-lapse. To compare the cell cycle stage for every nucleus as closely as

## BiFCo subnuclear locations in vivo

Immunofluorescence, immunoblotting, and numerous global in vitro genomic studies using different methods to identify cohesin's immunoprecipitated chromatin (ChIP) have mapped the distribution of cohesin across the genomes. These maps show a high concentration of this complex in telomeric and pericentromeric heterochromatin regions of the chromosomes in yeast and animal cells and in specific chromosome arm regions. These regions frequently coincide with convergent transcription sites in yeast (Blat & Kleckner, 1999; Tanaka et al, 1999; Tomonaga et al, 2000; Glynn et al, 2004; Gullerova & Proudfoot, 2008; Koch et al, 2008) and regions where CTCF zinc finger protein is located in mammals (Parelho et al, 2008), including humans (Holzmann et al, 2019). Thus, we tested whether the most prominent fluorescent foci observed by BiFCo correspond to these pools. We co-localized the BiFCo signal in vivo with well-established fluorescent markers of telomeres (Taz1 [Fig 3 inset I]), centromeres (Mis6 [Fig 3 inset II]), and the spindle pole body (Sid4 [Fig 3 inset III]), as centromeres remain clustered at the latter structure throughout most of the *Schizosaccharomyces pombe* cell cycle. The brightest BiFCo foci co-localized in all cells with either fluorescent marker signal. This confirms that it is possible to dynamically follow the accumulation of cohesin associated with centromeres and telomeres simultaneously (Fig 3), and it could serve as a simultaneous marker of these chromosomal regions in living cells during interphase.

In an asynchronous culture, 14.62% of cells had no evident foci (n = 130 non-sister nuclei). These likely correspond to cells in mitosis and early G1, where cohesin has been disassembled by Rad21 cleavage, and the amount is not detectable yet. We confirmed this assumption by addressing this figure in a *cdc10-129* temperature-sensitive mutant (see below). At the restrictive temperature, these cells arrest at the G1/S transition when a detectable amount of cohesin has already been reloaded onto the chromatin. As expected, we found only 3.2% of cells (n = 160) with no detectable BiFCo signal. These observations suggest that the BiFCo signal at these foci is cell cycle dependent. Apart from these sites, the system is sensitive enough to detect numerous smaller foci within the nuclei that likely correspond to accumulation loci along the chromosome arms (Fig 3 inset IV). Thus, the BiFCo pattern in vivo results strikingly similar to that found by chromosome spreads immunofluorescence against Rad21 in fixed cells, where soluble cohesin components are washed away, and only the chromatin-bound cohesin fraction remains visible (Ciosk et al, 2000; Schmidt et al, 2009).

## Total versus assembled cohesin components

Traditionally, tracking the assembly and disassembly dynamics of cohesin rings by fluorescence throughout the cell cycle or in experimental situations such as cohesin regulator mutant

backgrounds has been challenging. This is because only a proportion of its components form the complex, relative to the total amount of each component separately. Out of those, not all are bound to chromatin; and even out of the bound ones, not all are degraded in mitosis (Losada et al, 1998; Schmiesing et al, 1998; Darwiche et al, 1999; Tóth et al, 1999; Tomonaga et al, 2000; Weitzer et al, 2003; Schmidt et al, 2009; Holzmann et al, 2019). In mammalian cells, separase cleaves only 10% of cohesin complexes (Waizenegger et al, 2000; Koch et al, 2008), and *Drosophila* cells can tolerate an artificial decrease of 80% of total Rad21 levels before exhibiting cohesion defects (Carvalhal et al, 2018). The results suggest that most, if not all, of the Rad21-cleaved population, corresponds to the fraction bound to Psm1. It is unclear whether the quantum yield of the reconstituted molecule is comparable with that of the whole molecule, as there may be a difference between the biophysics of both. Thus, we assessed only qualitative differences between the fluorescence of the BiFCo system and traditional entire YFP fusions in the same time-lapse experiment with identical settings and signal drop proportion from G2 to M phase in each case (Figs 4A and S1 and Video 2). Because we had previously determined that the two daughter nuclei behave similarly regarding fluorescence recovery, we monitored, in this case, only one nucleus from each division to simplify the plotting. Interestingly, before mitosis, Psm1-YFP fluorescence signal is 3.8-fold higher than Rad21-YFP. In mitosis, the fluorescence levels of the total population of Psm1 are expected to drop by 50%, but Psm1 fluorescence drops only by 29.2%, suggesting a rapid import of this protein into the nucleus during anaphase that might compensate for fluorescence dilution at nuclear splitting. On the other hand, Rad21 average signal drops by 50.1% from the highest level in G2. Studies on fission yeast's cell lysate immunoblots have reported that less than 5% of Rad21 is cleaved in mitosis (Tomonaga et al, 2000). Therefore, further decay from the expected 50% may be too subtle to be detected as an average fluorescence of several nuclei. Alternatively, it might be compensated for with new protein import into the nucleus. In contrast to the whole fusions, the average BiFCo signal decreases by 68% in this period, making this unloading transition visually and quantifiably evident (Fig 4B). This drop likely accounts for cohesin disassembly, both dependent and independent of Rad21 proteolysis, although the latter seems to be a small fraction by immunoblotting and chromosome spread analyses (Schmidt et al, 2009). Thus, these data demonstrate that mitotic detachment of cohesin can be reliably monitored in living cells.

## Qualitative and quantitative BiFCo responses to cell cycle signaling

To confirm that the BiFCo signal is proportional to the amount of assembled complex and responds to cell cycle cues for cohesin loading and unloading dynamics, we impaired known regulators of these transitions and blocked the cell cycle in different stages. As

possible, we set out the first time-lapse of the oblong nucleus shape (beginning of anaphase B) as time 0. We used the average level of fluorescence of oblong nuclei minus the average background level at the same time-point to normalize fluorescence for each time-lapse (fNuclei-fBkd/fOblong each time-lapse-fBkd each time-lapse). Thus, normalized fluorescence of 1 represents the average intensity at the beginning of anaphase B. **(C, D)** Signal quantification (as in (C)) from section B over two consecutive mitotic events.

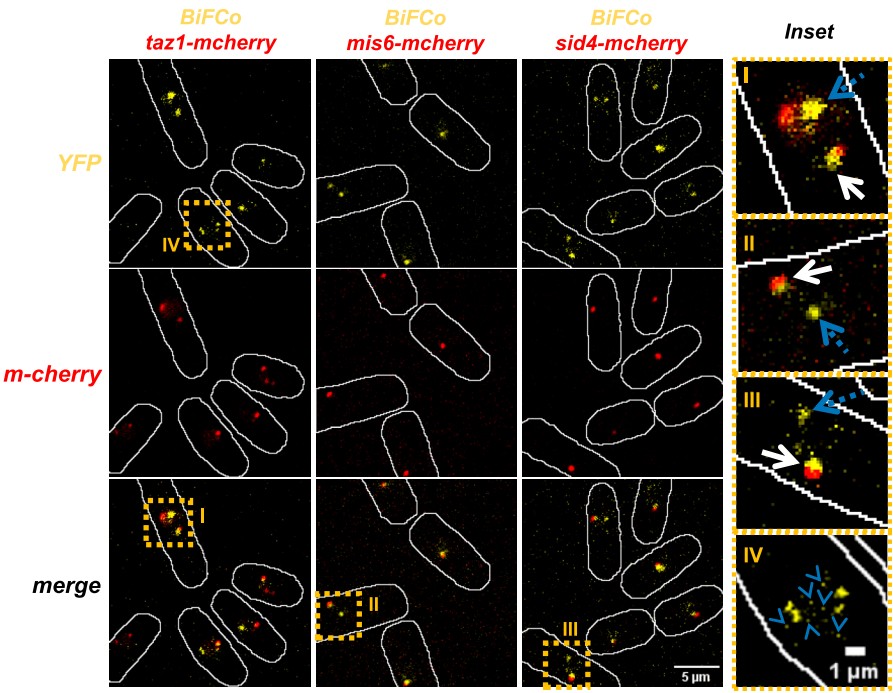

**Figure 3. Co-localization of BiFCo foci.**
We crossed our bimolecular fluorescent cohesin system into Taz1-mCherry, Mis6-mCherry, and Sid4-mCherry backgrounds to co-localize telomeres, kinetochores (centromeric region), and the spindle pole body, respectively. High-contrast images of maximal projections of 21 slices with a Z-step of 0.2 μm are presented. Single channel and merged images are shown. Scale bar: 5 μm. Inset enlargements of merged images are shown on the right (I, II, and III roman numbers). In all cases, some BiFCo main signal foci co-localize with either of the red markers (continuous white arrows), but others do not (blue dotted arrows). Bigger enlargement of high-contrast merged images (IV) also allows distinguishing numerous secondary foci (blue arrowheads). Scale bar: 1 μm. These data indicate that the more intense BiFCo signal corresponds to cohesin accumulation at centromeres and telomeres, and secondary foci represent assembled cohesin at chromosomal arm loci.

mentioned, cohesin release in mitosis mostly depends on Rad21 proteolysis, which breaks up the Psm1–Rad21 interaction (Cohen-Fix et al, 1996; Funabiki et al, 1996). To monitor BiFCo in a mutant background in which Rad21 cannot be degraded, we expressed the system in a *cut9-665* background, in which the APC/C is inactive at the restrictive temperature of 36°C and Rad21 cannot be degraded downstream (Samejima & Yanagida, 1994). After switching this strain to the restrictive temperature for 4 h, we mounted the cells for live microscopy and took time-lapse images every 3 min for two extra hours. As seen in Fig 5A and Video 3, cells with segregation defects after septation do not show a drop in the BiFCo signal when going through mitosis. Complementarily, we ensured that most of the observed BiFCo signal depends on the loading of the complex on chromatin in interphase. To test this, we observed the complex together with a red chromatin marker (Htt1-mRFP) in the temperature-sensitive *mis4-242* mutant background, which is defective in loading function and unable to maintain chromosome association at the restrictive temperature of 36°C (Tomonaga et al, 2000; Bernard et al, 2008). We observed a fluorescent signal similar to the wild-type background at the permissive temperature of 25°C, and even the foci of centromeres and telomeres were visible (Fig 5B, upper panels). However, when Mis4 function was impaired at 36°C, cohesion defects were evident as judged by the chromatin marker (red channel), and the BiFCo signal was reduced to hardly detectable levels (Fig 5B, lower right panel). These data together show that the BiFCo signal depends on the Mis4 loading function, and its severe drop in mitosis is a direct consequence of Rad21 proteolysis induced by the APC/C. The fact that the fluorescent signal almost disappears implies that the interaction in trans of both parts of YFP is reversible and responds to the natural process of cohesin disassembly.

We then investigated whether the bipartite fluorescent system could detect quantitative changes in cohesin ring assembly. Cohesin recruitment to chromatin in fission yeast begins in G1, and loading/unloading dynamics reach a steady state by this phase, similar to human cells (Guacci et al, 1997; Michaelis et al, 1997). Further accumulation and stabilization occur in the S phase and continue through to the G2 phase until mitosis (Uhlmann & Nasmyth, 1998; Bernard et al, 2008). Therefore, we arrested the cell cycle at different stages and quantified the BiFCo signal in 50 random nuclei for each stage. We assessed an asynchronous wild-type control, nitrogen-deprived cells (G1 arrest), *cdc10-129* mutant (START arrest), hydroxyurea (S phase arrest), and *cdc25.22* mutant (G2/M arrest). As expected, the asynchronous population, in which only about 10% of cells are in mitosis, showed no significant difference in the average signal intensity compared with G2-arrested cells. However, we detected a significant increase in the signal in asynchronous cells compared, respectively, with a G1 arrest after 4 h of nitrogen starvation (1.7-fold), a G1/S transition arrest (2.8-fold), and an early S phase block by hydroxyurea (4.1-fold) (Fig 6A and B). We also checked the total amount of Rad21 by Western blotting for each blockage used. Consistent with previous observations in fission yeast and human cells (Birkenbihl & Subramani, 1995; McKay et al, 1996), we observed Rad21 total protein levels peaking at the G2 phase (Fig 6C). Thus, even though total protein levels peak at G2, we did not observe a proportional increase in the BiFCo signal at this stage. These data support the idea that the amount of assembled cohesin is not proportional to the total amount of its components and show that BiFCo can serve as a quantitative marker of chromatin-loaded cohesin, with important implications for studying cohesin assembly and its coordination with the cell cycle.

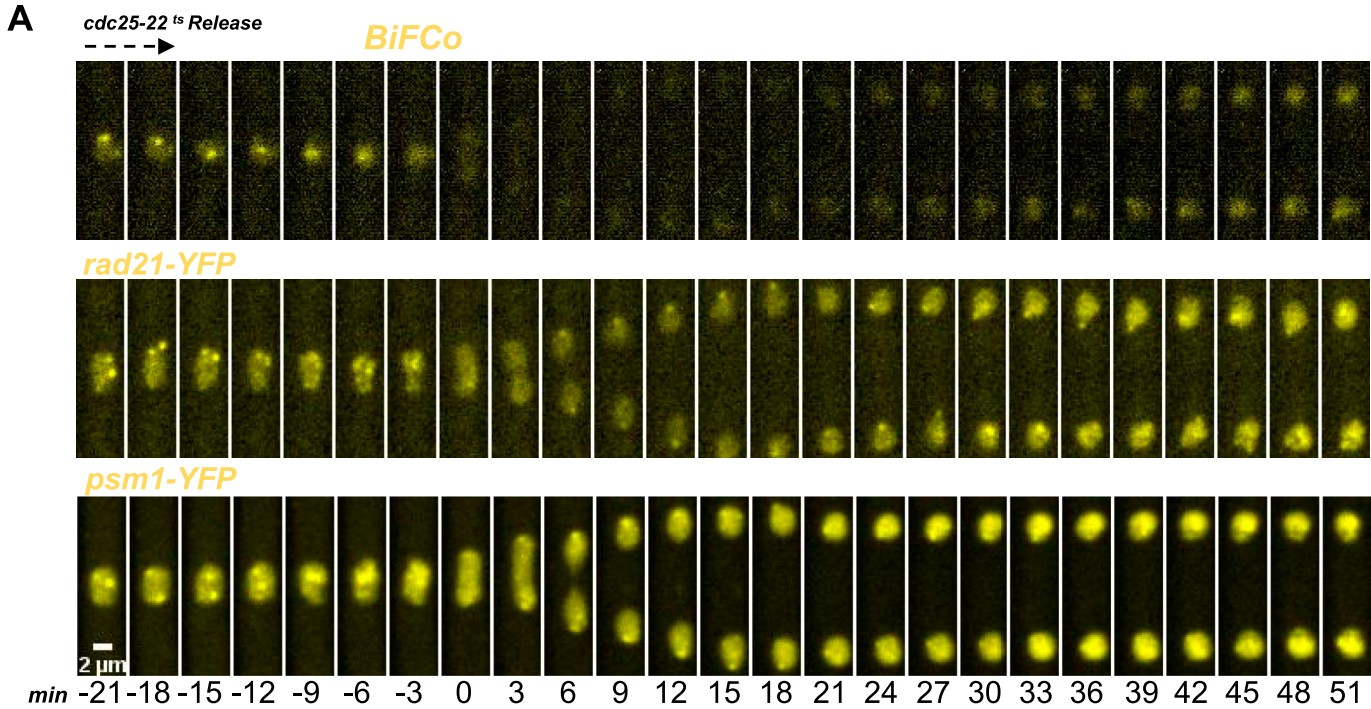

**A**

cdc25-22 ᵗˢ Release

BiFCo

rad21-YFP

psm1-YFP

2 µm

min -21 -18 -15 -12 -9 -6 -3 0 3 6 9 12 15 18 21 24 27 30 33 36 39 42 45 48 51

**B**

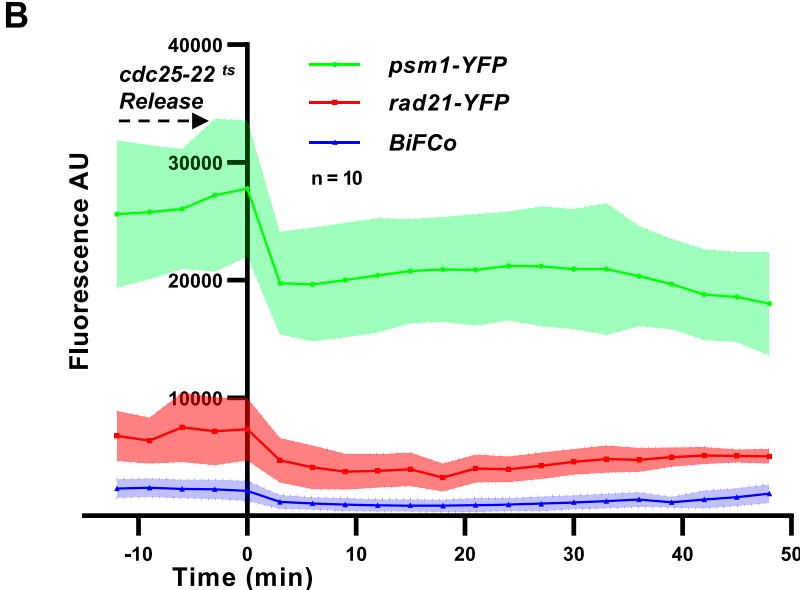

Figure 4. **Total versus assembled cohesin components.**
**(A)** Whole YFP fusions to Rad21 and Psm1, respectively, and the BiFCo strain, were mounted onto the same coverslip but spatially separated. This is achieved by sticking the cells to separate and confined soybean lectin drops. Time-lapse microscopy (traveling over the three strains at each time point) was performed under an identical illumination/capturing setup to compare fluorescent signals in cells getting into mitosis. Scale bar: 2 µm. Note that each strain's brightness and contrast levels presented in this figure are adjusted separately to distinguish nuclear foci in all strains. Real signal comparison with identical brightness and contrast levels can be seen in section B, Fig S1, and Video 1. **(B)** Average fluorescence intensity and SD (arbitrary units) are plotted. 10 nuclei for each strain were analyzed in the same cell cycle stage. The equivalent background area at each time-point for each cell was subtracted from the actual signal within the nucleus. It should be noted that the BiFCo system plotting appears to be flat because of the large increase in the scale of the arbitrary fluorescence units when whole individual fusion proteins are plotted along.

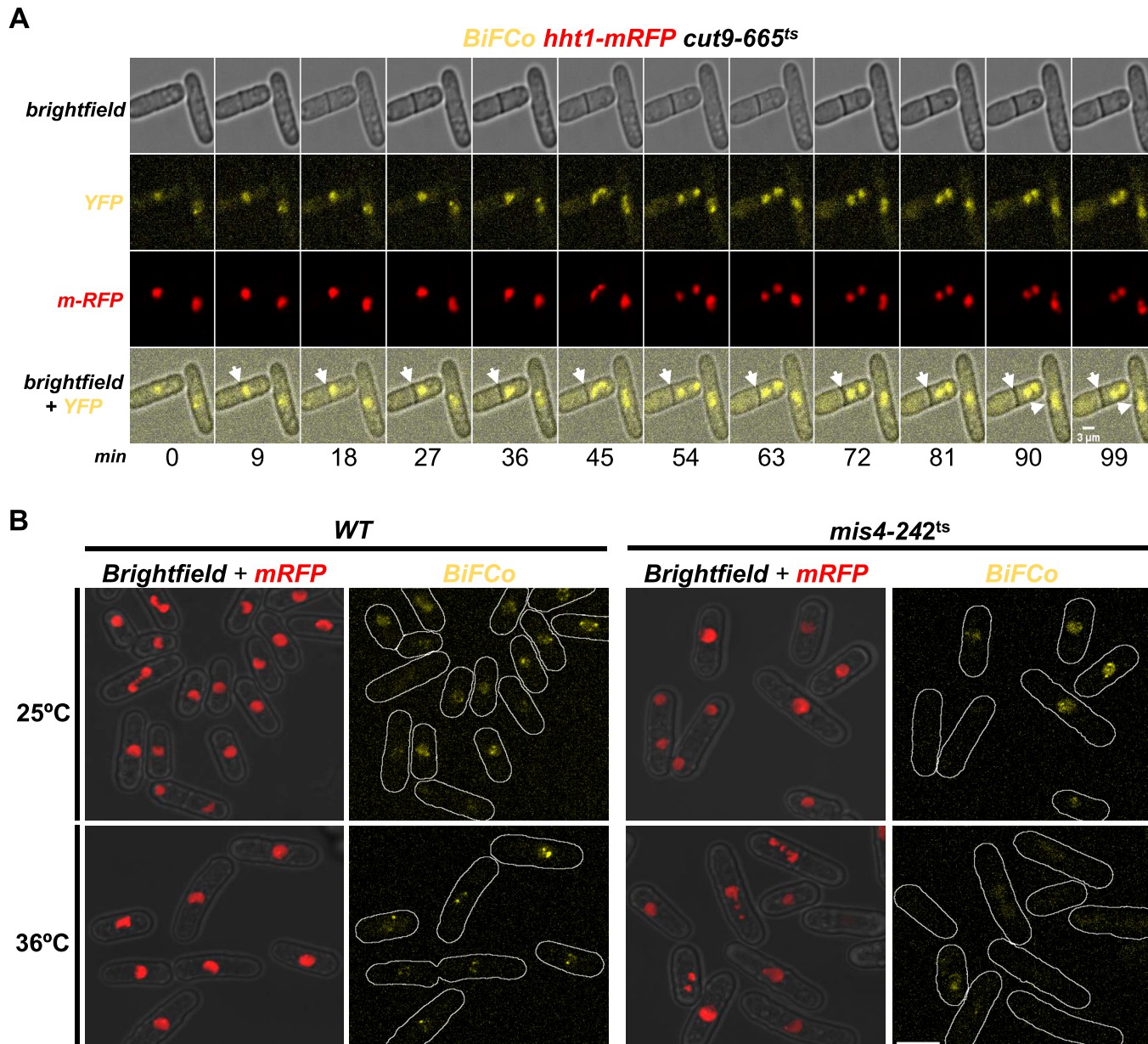

**Figure 5.  BiFCo signal depends on the Mis4 loader function, and the mitotic decay depends on Rad21 degradation.**
**(A)** The BiFCo tagging was expressed into the temperature-sensitive *cut9-665* mutant background. The APC/C function is abolished at high temperatures, so Rad21 is not cleaved. We cultured these cells until the early log phase at a permissive temperature of 25°C and shifted them to 36°C for 4 h before mounting them for time-lapse microscopy, keeping the sample at a high temperature under the microscope. Contrary to wild-type mitosis, the assembled complex's YFP signal does not decay in cells that go through anaphase. The time-lapse interval is indicated at the bottom: 9 min (Scale bar: 3 $\mu$m). **(B)** BiFCo complex and Htt1-mRFP (red chromatin marker) were expressed into *mis4*[+] and *mis4-242*[ts] backgrounds. Exponentially growing cultures at 25°C and shifted to 36°C for 4 h were processed for live cell microscopy. Maximal projections from 21 Z-stacks every 0.2 $\mu$m are shown for both strains and temperatures under identical settings. Scale bar: 7 $\mu$m. Fluorescence from the assembled complex is fully dependent on the Mis4 function.

## Discussion

The cohesin complex has been the subject of much attention over the last three decades. During this time, we have learned about its essential and evolutionarily highly conserved functions in eukaryotic genomes' architecture, expression, and segregation. Defects in the assembly of the complex's subunits and in the dynamics of loading and unloading onto chromosomes are common causes of genetic diseases in humans (Remeseiro et al, 2013). Therefore, studying these processes can help us understand and develop treatments for these diseases.

Fluorescence techniques, such as FRAP and FRET, have already revealed important structural features and dynamics of the cohesin complex in living cells (Gerlich et al, 2006; Mc Intyre et al, 2007;

**A**

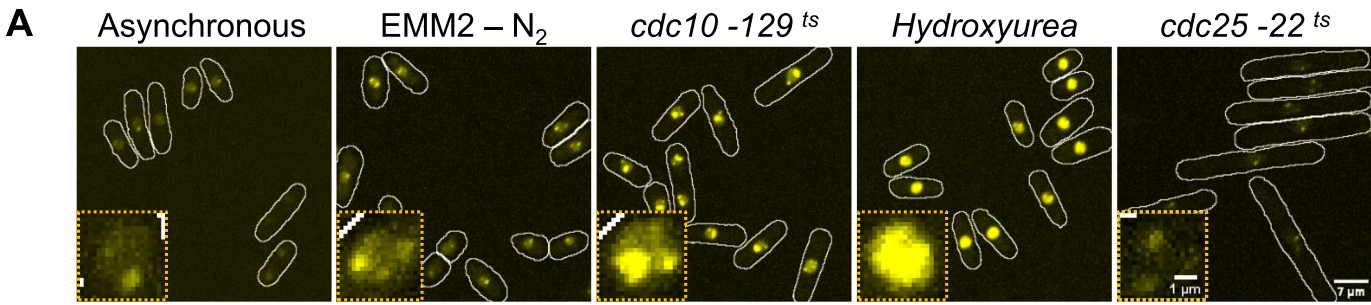

Asynchronous    EMM2 – N$_2$    *cdc10 -129* $^{ts}$    *Hydroxyurea*    *cdc25 -22* $^{ts}$

**B**

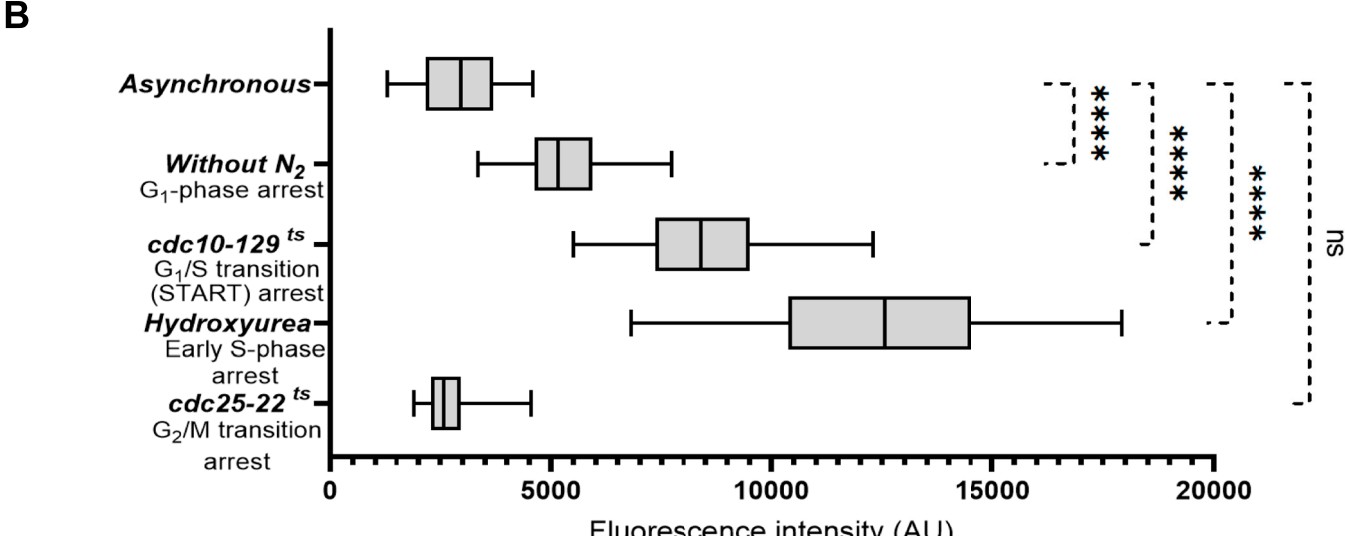

**C**

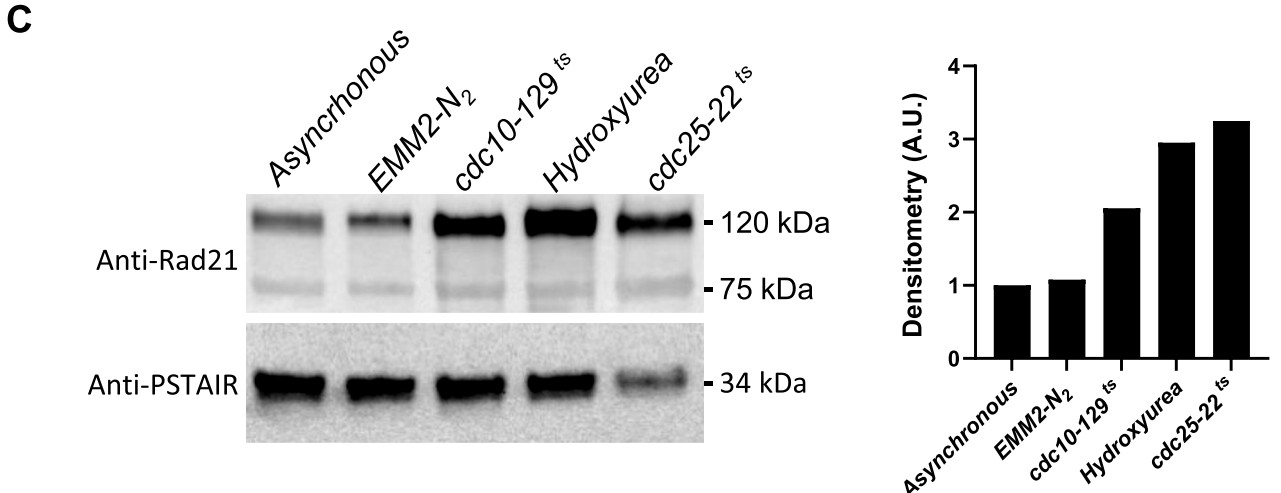

**Figure 6.  Assembled cohesin and Rad21 levels over the cell cycle.**
**(A)** BiFCo maximal projections (21 slices) from either asynchronous cells, nitrogen-starved cells for 4 h, *cdc10-129*-arrested cells at 36°C for 4 h, hydroxyurea-treated cells (10 mM, 4 h) or *cdc25-22*-arrested cells at 36°C for 4 h. Scale bar: 7 μm. The inset shows a representative nucleus enlargement in each case. Scale bar: 1 μm. Images are taken and adjusted identically for brightness and contrast. **(B)** Box and whisker distributions for signal quantification of 50 random nuclei in each condition. We used ordinary one-way ANOVA comparison against asynchronous control distribution to test statistical significance (GraphPad Prism 9.0.2 software. Dunnett's multiple comparisons test). Confidence interval *P* < 0.05. ns, non-significant. Asterisks indicate *P* < 0.0001. **(C)** Total Rad21 levels at each cell cycle stage. Total protein extracts were prepared from each cell cycle phase arrest. We used rabbit polyclonal anti-Rad21 antibody (A313; Antibodies.com) to assess total Rad21 protein by Western blot and monoclonal anti-PSTAIR antibody (P7962; Sigma-Aldrich) for loading control. Densitometry analysis was performed with Image Lab 6.0.1 software. The graph on the right shows protein levels standardized to respective loading controls and relative to the asynchronous culture.

Holzmann et al, 2019). However, technical limitations in sensitivity and temporal resolution have made detecting transient cohesin structural changes and long-term loading and unloading transitions impossible. In addition, complete FP fusions can make it difficult to distinguish the signal of total components from the assembled fraction. Most cohesin loading, structure, and regulation studies have been carried out in vitro, in reconstituted reactions or by analyzing cell lysates or fixed chromosome spreads using immunofluorescence (Bernard et al, 2008; Xiang & Koshland, 2021).

In this study, we have developed a proof-of-principle BiFCo, which can selectively discriminate the assembled complex within a physiological context, excluding the soluble fraction of individual proteins present in living cells. Our findings show that separate fusions of the two complementary parts of vYPF to Psm1 and Rad21, respectively, only produce fluorescence when these proteins interact as part of the complex. This is consistent with in vitro cohesin-associated chromatin immunoprecipitation analyses (ChIP-seq, ChIP–chip), fixed chromosome spreads, and FRET studies (Blat & Kleckner, 1999; Tanaka et al, 1999; Mc Intyre et al, 2007; Schmidt et al, 2009). The BiFCo signal is detected at major sub-nuclear foci in interphase, corresponding to centromeric and telomeric regions of the chromosomes. In addition, the sensitivity of this system allows for detection at minor foci, which likely correspond to clusters along the chromosome arms.

Bimolecular complementation is particularly useful for studying dynamic protein interactions in vivo within the same cell over time. However, it faces two potential limitations. Firstly, the folding time of the two parts of YFP fused to their respective baits and the time needed for the association of fluorescent fragments and chemical reactions leading to fluorophore production may be limiting when the protein turnover in the given biological process is very fast. Secondly, the recomposition of the two parts of YFP in trans becomes very stable once produced, making it difficult to study naturally reversible protein interactions in vivo (Kerppola, 2009).

However, in the experimental system presented here, these limitations are inherently overcome by the biology of cohesin itself. A large pool of free components, in addition to preassembled complexes, suggests that the cell has a natural reservoir of complexes and soluble partners ready to be loaded or coupled de novo to ensure rapid reloading in the G1 phase. Thus, BiFC-tagged proteins should have sufficient time for synthesis and folding before they are assembled in the next cycle. Our analysis illustrates this by the rapid onset of fluorescence recovery from the lowest intensity time-lapse, which occurs in the next 20 min, coincident with the wild-type G1 phase.

In *Saccharomyces cerevisiae*, once separase cleaves SCC1/Rad21, the N-end rule pathway rapidly degrades the resulting C-terminal fragment, and it causes lethality when artificially stabilized (Rao et al, 2001). The cleavage recognition sites in SCC1/Rad21 are similar between budding and fission yeast, and the amino acids left at the N-terminus of the resulting fragments are compatible with the degradation system by the N-end rule pathway (Varshavsky, 1996; Tomonaga et al, 2000). We did not observe any sign of anaphase progression defects in the BiFCo strain, which the altered half-life of the Rad21-fragmented products could cause. Therefore, it is reasonable to consider that this degron system also operates in *S. pombe*, and thus the YFP moiety fused to the Rad21 C-terminal

fragment is quickly degraded after cleavage. Even if the interaction of the two parts of YFP were very stable, in this case, it should not pose a problem because we are monitoring a stable interaction that occurs only once between two partners and is naturally destroyed to reset the system at each cell cycle.

An interesting observation after two complete mitotic cycles in this setup is that after long-term exposure (8 h) of cells to the excitation laser, the G2 intensity levels and the fluorescence recovery rate in the second cycle are very similar to the first one. This suggests that the separate parts of YFP do not undergo significant photobleaching, which is advantageous for following the complex over several cell cycles or in model organisms with prolonged cell cycles.

Using full YFP tagging, the average nuclear fluorescence suggests that the total levels of Psm1 in the G2 phase are 3.8 times higher than Rad21 levels. Although not in the scope of this work, this roughly 4:1 stoichiometry represents a considerable difference compared with that observed in budding yeast, where equal protein levels are detected by Western blot (Mc Intyre et al, 2007).

Tracking loading and unloading of complete fusions throughout the cell cycle is not very apparent in either yeast because of the large background of soluble protein within the nucleus. However, the BiFCo complex behaves physiologically as expected, as the loading depends on the Mis4 loader function, and the signal abruptly decays by almost 70% in mitosis, making the cohesion dissolution detectable in living cells. We show that the separase-mediated cleavage of Rad21 kleisin causes this signal drop because this degradation is not observed when APC/C function is inhibited in the *cut9.665ts* mutant background.

Nevertheless, detectable basal signal levels remain during wild-type anaphase, which may correspond to clustered rings but chromatin-independent, single-chromatid-assembled complex and/or non-cohesive cohesin clusters.

The signal decay to almost undetectable levels in the *mis4* defective background may be unexpected, as it has been previously reported by co-immunoprecipitation assays in budding yeast that cohesin complexes remain assembled even in the absence of SCC2 function (*sp*Mis4 ortholog) (Ciosk et al, 2000). However, likely, cohesin complexes that remain assembled but unloaded from the chromatin in the absence of *mis4* function do not generate a detectable fluorescent signal because of a dilution effect within the nucleoplasm. In contrast, co-immunoprecipitation blots would easily detect this association as described in Ciosk et al, 2000. This finding is consistent with previous experiments performed in fission yeast, the same model used in this work. The authors monitored the signal of Rad21GFP complete fusion in the *mis4-242* temperature-sensitive background at permissive and restrictive temperatures. Upon inactivation of the *mis4* function, they observed that not only the signal from the centromere and telomere foci but also a significant portion of the fluorescent background within the nucleus disappears, keeping comparable Rad21 total protein levels to permissive conditions (Tomonaga et al, 2000).

In yeast, cohesin is recruited to chromatin and stabilized throughout the cell cycle comprising G1 and S phases. Therefore, we asked whether loading signals would remain active in cells arrested over these periods. We observed that a short block in G1 by removing the nitrogen source for only 4 h resulted in a 1.7-fold increase

in G2 signal levels. Interestingly, the assembly/stabilization process appears to be continuous and cumulative because it keeps increasing up to 2.8-fold if the block occurs slightly later in the G1/S transition (known as START in yeast and "restriction point" in mammals) and up to fivefold if the cell is arrested early in the S phase by the addition of hydroxyurea (HU).

For the interpretation of HU-mediated arrest, it must be considered that HU also induces DNA damage and activates the DNA replication checkpoint. Some cohesin mutants are hypersensitive to several DNA-damaging agents, including hydroxyurea, and this drug induces SCC1/Rad21 phosphorylation (Kim et al, 2002; Schär et al, 2004). Therefore, in addition to the loading process that may be maintained while the cell transits through the S phase, regardless of whether the actual progression is blocked, activation of the replication checkpoint may also contribute to the signal increase.

The BiFCo signal increase in this experiment suggests that the cell loads an amount of cohesin proportional to the temporal duration of the G1/S window when the loading signaling is active. Therefore, the chromatin will reach the G2 phase with that cohesin load. However, suppose the cell is blocked in the G1 or S phase, in that case, the cell cycle-coupled loading signals will likely remain active for the duration of the block, and it might be expected to find more cohesin assembled/accumulated on chromatin because of the abnormal duration of these cell cycle phases.

Regarding the relationship with the amount of total protein in Rad21, it has been shown in humans that the expression increases, especially during the S phase, and reaches maximum mRNA levels in G2 (McKay et al, 1996). A very similar regulation was reported in *S. pombe*, where mRNA levels appear to peak in the S phase, but the highest total protein levels are found in G2-arrested cells (*cdc13-117* block) (Birkenbihl & Subramani, 1995). Consistently, we found increased levels of total Rad21 protein by Western blotting from G1-, S-, and G2-arrested cells (Fig 6C). Interestingly, despite having maximum total protein levels in G2, we did not observe a proportional increase in the BiFCo signal at this stage. This reinforces the idea that the amount of assembled cohesin is not proportional to the total amount of its components. These data suggest that BiFCo can reveal quantitative cohesin assembly phenotypes in a dynamic range of at least fivefold over the wild-type signal in G2.

To our knowledge, the work presented here is the first application of a split-FP complementation setup to visualize cohesin and track assembly/disassembly dynamics in time-lapse microscopy within a living cell over a whole cell cycle. We believe this system could be expanded and diversified by fusing the complementary parts of suitable fluorescent proteins to different domains and/or proteins forming the complex and to their regulators. Integrating BiFCo with diverse mutant genetic backgrounds could also help design high-content screens and experiments to answer important questions within a living cell context about the intricate functions, dynamics, structural biology, and recent loop extrusion models of cohesin (reviewed in Higashi and Uhlmann [2022]). This approach could also be tested in other structurally similar SMC complexes such as condensin and SMC5/6 (Anderson et al, 2002; Nasmyth & Haering, 2005; Yatskevich et al, 2019; Sole-Soler & Torres-Rosell, 2020; Kim, 2021). Moreover, the high evolutionary conservation of the structure and regulation of these complexes in eukaryotic organisms and the advent of CRISPR-Cas editing systems make it

possible to easily implement this technique in other biological models from yeast through to human cultured cells.

# Materials and Methods

### Media and growth conditions

As described by Moreno et al (1991), standard fission yeast growth media and molecular biology approaches were used throughout the study. Sporulation agar (SPA) was employed for mating and sporulation, whereas segregation analyses were performed by tetrad pulling using a Singer MSM 400 automated dissection microscope from Singer Instruments. Cells were cultivated to the mid-log phase in EMM2 media for spot tests, cell count/ml was determined in a Neubauer chamber, and matching dilutions were made for all cultures. Serial fivefold dilutions were then plated onto solid media.

### Gene tagging

Gene tagging followed the protocol described in Bahler et al (1998) utilizing the pF6aMX6 plasmid series. After transformation, potential tagged strains were assessed via PCR to confirm integration at the anticipated loci.

### Immunoblotting

The cells were cultured to the exponential phase and then subjected to nitrogen-depleted media, high temperature or hydroxyurea treatment, as indicated in the text. Pellets were washed and suspended in 20% TCA, followed by the addition of 200 $\mu$l chilled glass beads. Cell disruption was performed in a Fastprep device (116005500; MP Biomedicals) through three cycles at a maximum power of 20 s with 1 min on ice in between. Next, 150 $\mu$l of 20% TCA was added, and the extracts were transferred to a new tube containing 1 ml of 5% TCA and then centrifuged at 2,500$g$ at 4°C for 5 min. The pellet was resuspended in 70 $\mu$l of 1 M Tris ClH pH 8, followed by the addition of 70 $\mu$l of 2× SDS loading buffer. The samples were boiled at 95°C for 5 min and centrifuged at 17,000$g$ for 5 min at room temperature to clarify the extracts. Next, 5–10 $\mu$l of the sample was loaded onto a 10% SDS–PAGE mini gel. After transferring to a nitrocellulose membrane, Rad21 was detected using a rabbit polyclonal anti-Rad21 antibody (1:2,000, A313; Antibodies.com) and a secondary anti-rabbit IgG-peroxidase antibody (1:5,000, A8275; Sigma-Aldrich). Monoclonal anti-PSTAIRE antibody (1:2,000, P7962; Sigma-Aldrich) was used for loading controls with a secondary anti-mouse IgG-peroxidase antibody (1:5,000, A5278; Sigma-Aldrich). The SuperSignal West Fento kit (34095; Thermo Fisher Scientific) was employed to develop the blot. Image documentation was obtained using the ChemiDoc MP Imaging System (12003154; Bio-Rad).

### Fluorescence microscopy

Fluorescence images were acquired from cells in the exponential growth phase using a Zeiss Axio Observer 7 inverted microscope equipped with Zeiss Plan-Apochromat 63×/1.40 Oil DIC and Alpha

Plan-Apochromat 100×/1.46 Oil DIC lenses coupled with a Spinning Disk Confocal Yokogawa CSU-W1 head. The system was equipped with solid-state excitation lasers (50 mW 515 nm [YFP] and 561 nm [RFPs]) and a DBP emission filter (527/54 + 645/60) from 3i (Intelligent Imaging Innovations). Device control and image acquisition were performed using SlideBook 6 software, and the cells were mounted in a Biopthec FCS2 chamber to enable temperature control during microscopy. When comparing fluorescent signals between two or more strains, all strains were placed next to each other but physically separated onto *Glycine max* lectin (L1395; Sigma-Aldrich) delimited drops on the coverslip. All images were processed using open-source Image J (Fiji) software (Schneider et al, 2012), and presented images correspond to maximal or sum projections as indicated in each figure legend.

# Supplementary Information

# Acknowledgements

We thank Dr. Iain Hagan for providing the BiFC tagging plasmids and tagged strains and Dr. Mitsuhiro Yanagida, Dr. Paul Nurse, and Dr. Julia Cooper for the mutant and tagged strains. Katherina García for her assistance in the advanced microscopy facility, Laura Tomas in the proteomics facility, Modesto Berraquero and Sergio Villa for their technical advice, and Victor Carranco for his excellent lab technical support and digital design advice. We also thank Dr. Ignacio Flor-Parra for providing comments on the manuscript. This work was funded by Programa operativo FEDER/Junta de Andalucía 2014–2020, Objetivo específico 1.2.3. Grant UPO-1381219 to VA Tallada and Ministerio de Ciencia e Innovación Grant PID2019-111124GB-I00 to J Jiménez. E González-Martín is funded by Pablo Olavide University grant "Ayudas para Investigación Tutorizadas" modalidad B-2 (Rf.: PPI2104).

## Author Contributions

E González-Martín: formal analysis, validation, investigation, visualization, and writing—review and editing.
J Jiménez: conceptualization, resources, supervision, and funding acquisition.
VA Tallada: conceptualization, supervision, funding acquisition, validation, methodology, and writing—original draft, review, and editing.

## Conflict of Interest Statement

The authors declare that they have no conflict of interest.

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
