## [Reviewer comments · Life Science Alliance]

BiFCo: Visualizing cohesin assembly/disassembly cycle in living cells

Emilio González-Martín, Juan Jiménez and Víctor Tallada

DOI: 10.26508/lsa.202301945

Corresponding author(s): Dr. Víctor A. Tallada (Universidad Pablo de Olavide)

Review timeline:

Submission Date:	2023-01-24
Editorial Decision:	2023-02-17
Revision Received:	2023-03-31
Editorial Decision:	2023-04-11
Revision Received:	2023-04-13
Accepted:	2023-04-17

Scientific Editor: Eric Sawey

Transaction Report:

No Peer Review Process File is available with this article, as the authors have chosen not to make the review process public in this case.

Re: Life Science Alliance manuscript #LSA-2023-01945-T

Dr. Víctor A. Tallada
Universidad Pablo de Olavide
CABD
Ctra Utrera Km1
Seville 41013
Spain

Dear Dr. Tallada,

Thank you for submitting your manuscript entitled "BiFCo: Visualising cohesin assembly/disassembly cycle in living cells" to Life Science Alliance. The manuscript was assessed by expert reviewers, whose comments are appended to this letter. We invite you to submit a revised manuscript addressing the Reviewer comments.

I appreciate that this work is presented as a Methods paper, so some of Reviewer 1's early points are beyond the scope of this work. However, the points made about mechanistic insight can be addressed through added discussion, and any further insight you may be able to provide. I would be happy to evaluate a revision plan so that you can spend your time on the revision efficiently.

Thank you for this interesting contribution to Life Science Alliance. We are looking forward to receiving your revised manuscript.

Sincerely,

- A letter addressing the reviewers' comments point by point.
- An editable version of the final text (.DOC or .DOCX) is needed for copyediting (no PDFs).
- High-resolution figure, supplementary figure and video files uploaded as individual files: See our detailed guidelines for preparing your production-ready images, <https://www.life-science-alliance.org/authors>
- Summary blurb (enter in submission system): A short text summarizing in a single sentence the study (max. 200 characters including spaces). This text is used in conjunction with the titles of papers, hence should be informative and complementary to the title and running title. It should describe the context and significance of the findings for a general readership; it should be written in the present tense and refer to the work in the third person. Author names should not be mentioned.
- By submitting a revision, you attest that you are aware of our payment policies found here: <https://www.life-science-alliance.org/copyright-license-fee>

B. MANUSCRIPT ORGANIZATION AND FORMATTING:

RE: Life Science Alliance Manuscript #LSA-2023-01945-TR

Dr. Víctor A. Tallada
Universidad Pablo de Olavide
CABD
Ctra Utrera Km1
Seville 41013
Spain

Dear Dr. Tallada,

Thank you for submitting your revised manuscript entitled "BiFCo: Visualizing cohesin assembly/disassembly cycle in living cells". We would be happy to publish your paper in Life Science Alliance pending final revisions necessary to meet our formatting guidelines.

-please add a callout in the text for Supplementary Video 3

A. FINAL FILES:

-- Summary blurb (enter in submission system): A short text summarizing in a single sentence the study (max. 200 characters including spaces). This text is used in conjunction with the titles of papers, hence should be informative and complementary to the title. It should describe the context and significance of the findings for a general readership; it should be written in the present tense and refer to the work in the third

person. Author names should not be mentioned.

B. MANUSCRIPT ORGANIZATION AND FORMATTING:

Sincerely,

RE: Life Science Alliance Manuscript #LSA-2023-01945-TRR

Dr. Víctor A. Tallada
Universidad Pablo de Olavide
CABD
Ctra Utrera Km1
Seville 41013
Spain

Dear Dr. Tallada,

Thank you for submitting your Methods entitled "BiFCo: Visualizing cohesin assembly/disassembly cycle in living cells". It is a pleasure to let you know that your manuscript is now accepted for publication in Life Science Alliance. Congratulations on this interesting work.

DISTRIBUTION OF MATERIALS:

Again, congratulations on a very nice paper. I hope you found the review process to be constructive and are pleased with how the manuscript was handled editorially. We look forward to future exciting submissions from your lab.

Sincerely,
